# Cell-Type Resolved Insights into the Cis-Regulatory Genome of NAFLD

**DOI:** 10.3390/cells11050870

**Published:** 2022-03-03

**Authors:** Trine V. Dam, Nicolaj I. Toft, Lars Grøntved

**Affiliations:** Department of Biochemistry and Molecular Biology, University of Southern Denmark, DK-5230 Odense, Denmark; trinevd@bmb.sdu.dk (T.V.D.); nicolajit@bmb.sdu.dk (N.I.T.)

**Keywords:** NAFLD, NASH, liver, chromatin, transcription factor, gene regulatory network, cis-regulatory region, single-cell analysis

## Abstract

The prevalence of non-alcoholic fatty liver disease (NAFLD) is increasing rapidly, and unmet treatment can result in the development of hepatitis, fibrosis, and liver failure. There are difficulties involved in diagnosing NAFLD early and for this reason there are challenges involved in its treatment. Furthermore, no drugs are currently approved to alleviate complications, a fact which highlights the need for further insight into disease mechanisms. NAFLD pathogenesis is associated with complex cellular changes, including hepatocyte steatosis, immune cell infiltration, endothelial dysfunction, hepatic stellate cell activation, and epithelial ductular reaction. Many of these cellular changes are controlled by dramatic changes in gene expression orchestrated by the cis-regulatory genome and associated transcription factors. Thus, to understand disease mechanisms, we need extensive insights into the gene regulatory mechanisms associated with tissue remodeling. Mapping cis-regulatory regions genome-wide is a step towards this objective and several current and emerging technologies allow detection of accessible chromatin and specific histone modifications in enriched cell populations of the liver, as well as in single cells. Here, we discuss recent insights into the cis-regulatory genome in NAFLD both at the organ-level and in specific cell populations of the liver. Moreover, we highlight emerging technologies that enable single-cell resolved analysis of the cis-regulatory genome of the liver.

## 1. Introduction

Non-alcoholic fatty liver disease (NAFLD) is driven by a western lifestyle-associated increased flux of lipids and carbohydrates into the liver and remodeling of the gut-microbiome [1,2,3]. The disease ranges from mild steatosis (non-alcoholic fatty liver, NAFL) to severe steatosis with inflammation and fibrosis (non-alcoholic steatohepatitis, NASH) [2]. Failure of the parenchymal hepatocytes to cope with the chronically high lipid load is a critical event in NAFLD pathology, and the consequential lipotoxicity triggers a cascade of pathogenic events in the non-parenchymal cell populations such as the activation of Kupffer cells (KCs), immune cell infiltration, liver sinusoidal endothelial cell (LSEC) capillarization, hepatic stellate cell (HSC) activation, and epithelial ductular reaction comprising activation, proliferation, and potential transdifferentiation amongst cholangiocytes, bipotent biliary progenitor cells or hepatocytes, or a combination of these [2,4,5,6,7,8]. This causes dramatic alterations of tissue architecture and function that ultimately may lead to NASH and can result in cirrhosis and liver failure [2,9]. These cellular behaviors are in part orchestrated by complex gene regulatory networks that change the gene expression signature of individual cells through extensive chromatin remodeling events, altered transcription factor (TF) binding patterns, and histone modifications [10,11,12]. Despite its high prevalence (~25% of all adults), no FDA approved NAFLD therapy exists, and simply regaining liver function largely depends on early diagnosis to motivate lifestyle changes when the disease is still at the mild end of the severity spectrum [13]. However, early diagnosis is challenged by the asymptomatic nature of NAFLD leading to the patients only being diagnosed when the disease starts to compromise tissue function substantially. In addition, the early detection of NAFLD is complicated by the invasiveness of existing diagnosis approaches involving liver biopsy procedures combined with surrogate clinical parameters such as circulating levels of alanine aminotransferase activity and elastography [2,14]. Taken together, there is a great need to establish new NAFLD diagnosis and treatment approaches and these will require deeper insight into disease mechanisms. Focusing on gene regulatory mechanisms, this review discusses using genomics approaches to identify the cis-regulatory elements (also referred to as the cistrome) in the individual cell types of the liver and highlights studies that have applied these in a NAFLD context. In addition, we feature emerging single-cell epigenetic technologies used to study cell-type-resolved gene regulatory networks in NAFLD, which we foresee will open up important new avenues of diagnosis and therapy possibilities in the coming years.

## 2. Identification of Cis-Regulatory Regions in the Diseased Liver

The cistrome encompasses the set of cis-regulatory elements that contribute to the tight control of gene transcription in a cell. These elements are occupied by TFs and co-regulators to keep them in a certain regulatory state; either repressive, active, or dormant [15,16]. Regulatory regions occupied by TFs are generally more accessible compared to silenced sites organized in less accessible chromatin. These different chromatin conformations can be probed by endonucleases or transposases such as DNaseI or Tn5 leveraged in methodologies like DNase-seq and Assay for transposase-accessible chromatin (ATAC-seq) (Figure 1A) [17]. Moreover, the activity state of cis-regulatory elements is associated with specific histone modifications and in combination with the three-dimensional structure of chromatin this defines nanoscale molecular environments that govern the transcription of target genes (Figure 1A). Specifically, accessible regions surrounded by deacetylated H3K27 and monomethylated H3K4 (H3K4me1) are considered dormant cis-regulatory regions poised for activation, whereas H3K27Ac and H3K4me1/2 mark their active state [16,18]. Thus, chromatin immunoprecipitation (ChIP-seq) against histone modifications can probe these regulatory states genome-wide, while Hi-C methodology provides insight into genes regulated by the cis-regulatory elements by capturing their physical interactions [19] (Figure 1A). In addition, active cis-regulatory elements are transcribed and release enhancer RNAs (eRNAs) [20]. These can be captured and quantified by methods such as Global Run-On (GRO-seq) and can thus profile the activity of cis-regulatory elements [21]. Taken together, cis-regulatory elements of the genome can be identified by several sequencing-based methodologies, and downstream bioinformatic analysis can predict TFs controlling accessibility and activity of these regions, ultimately leading to the unraveling of complex gene regulatory networks that define cellular states (Figure 1B) [22]. Over the years, several studies have characterized NAFLD at a bulk tissue level using one or more of the above-mentioned approaches, producing datasets reflecting the average chromatin remodeling processes of all cells composing the liver [23,24,25,26,27,28]. As such, mainly hepatocyte cis-regulatory regions (>60% of all liver cells) are captured in healthy conditions, whereas immune cells likely contribute to datasets generated from inflamed and fibrotic liver tissue. Despite this bias, these initial studies have provided the field with invaluable insights into the fundamental gene regulatory mechanisms in mouse NAFLD.

### Hepatic Chromatin Remodeling Induced by NAFLD

Assessing hepatic chromatin accessibility and histone acetylation in diet-induced obesity (DIO) mouse models have revealed dramatic changes to chromatin organization in the liver during NAFLD development [23,26,27,28]. Although, the reversibility of these changes upon clearance of steatosis is a matter of debate [23,26]. Interestingly, DIO remodels circadian activity of hepatic cis-regulatory elements [29] and rewires physical interactions between cis-regulatory regions and gene promoters [28]. Thus, NAFLD is generally associated with dramatic alterations in the hepatic chromatin landscape at the overall tissue level. Mining sequence similarities of the identified cis-regulatory regions has provided insights into specific TFs that are involved in DIO-mediated NAFLD development, including HNF4α, C/EBPα, SREBP-1c, and PPARα [23,26,28].

Downregulation of HNF4α in obese mice and NASH patients [28,30] has been linked to NASH pathogenesis, supported by observations of alleviated fibrosis in CCl_4_-induced NASH models upon induced HNF4α overexpression [31,32,33]. Intriguingly, the occupancy of HNF4α at regulatory regions is redistributed in DIO [28], where the underlying mechanism may be linked to cooperation with C/EBPα [34]. C/EBPα regulates hepatic lipogenesis and shows increased DNA binding activity in NAFLD patients [35,36,37]. Thus, attenuating C/EBPα expression reduces lipogenic gene expression and steatosis in obese mouse models [38,39], highlighting its central role in NAFLD. Lipogenesis is further regulated by SREBP-1c [40], which activates gene expression in this pathway in response to insulin signaling [41,42]. In an NAFLD context, both SREBP-1c expression and the circadian amplitude increase [29,43,44,45], and its DNA binding motif is enriched in sites gaining circadian activity in DIO. Confirming its role in NAFLD pathogenesis, hepatocyte-specific SREBP-1c overexpression leads to induced lipogenesis and hepatosteatosis [46], while the knockout (KO) of SCAP, a protein required for SREBP-1c activity [47], inhibits the expression and rhythmicity of lipogenic genes [29]. Interestingly, in alignment with SREBP-1c rhythmicity, PPARα shows induced circadian expression amplitude in DIO and enriched motif activity in DIO-induced circadian cis-regulatory regions [29]. PPARα activity is associated with fatty acid oxidation (FAO), which has been reported to be upregulated in NAFLD patients [48]. This suggests simultaneous activation of lipogenesis and lipolysis in the steatotic liver and indicates a protective PPARα function, supported by PPARα KO studies showing the exacerbation of NAFLD progression [49,50] and fenofibrate (PPARα agonist) studies showing reduced steatosis [51,52,53]. Counterintuitively, hepatocyte-specific SCAP-KO decreases FAO, without significantly altering PPARα levels [29]. However, studies suggest that SREBP-1c is implemented in the synthesis of proteins involved in PPARα ligand generation [29], and thus controls the temporal PPARα occupancy in NAFLD, emphasizing the complex cooperation that exists between TFs.

## 3. Strategies to Study Cis-Regulatory Regions in Individual Cell Populations of the Liver

NAFLD is driven by gene regulatory changes in distinct cell type populations of the liver, emphasizing the importance of studying the disease at cell type resolution. This can be approached by the enrichment of specific cell types prior to the cistromic analysis (Figure 2A) [12,54,55,56,57]. Enzyme-based tissue dissociation is a commonly used approach to isolate whole cells from fresh tissues of rodent disease models and human biopsies (Figure 2A, right and Figure 2B). Specific cell population can subsequently be enriched by fluorescence-activated cell sorting (FACS) or affinity purification against cell specific proteins/markers, labelled cells, or both (Figure 2B). This offers a range of possibilities to sort for specific cell populations (Figure 2C), although its workflow has been shown to impact cellular processes considerably [58]. Furthermore, some cell types dissociate under different experimental conditions, challenging the analysis of several cell types from the same biological replicate [58]. Lastly, the isolation of intact cells taken from frozen tissue is not possible, which reduces flexibility and complicates the analysis of cryopreserved tissues and biopsies (Figure 2C). In contrast, nuclei can be isolated from fresh or frozen tissue through more simple procedures such as Dounce homogenization which induces mechanical cell lysis (Figure 2A, left and Figure 2C). Combined with cell specific nuclei tagging strategies, this allows the cistromic analysis of specific cell populations (Figure 2B). However, native nuclei are fragile to handle, the exclusion of cytosolic mRNAs can leave out relevant biological information and certain immune cell populations have been shown to be underrepresented in the released nuclei fraction [58]. Both nuclear and cellular isolation are applicable for subsequent analysis of chromatin accessibility and histone modifications associated with active cis-regulatory regions (Figure 2A). As discussed below, the isolation strategies are also aligned with single-cell/nucleus analysis.

To enrich for a given cell type from a given tissue, FACS is widely used and applicable to both cells and nuclei. However, sorting occurs under stress-inducing conditions far from the natural tissue environment [59]. Moreover, pre-knowledge of cell type marker genes is required as cellular detection relies on fluorophore-tagging (often antibody-based) and analysis of rare cell types is practically challenged by long sorting times. The latter issue can to some extend be circumvented by enriching for rare cell types prior to sorting, either by the removal of abundant cell populations using cell surface markers or by gradient centrifugation, as exemplified by studies of non-parenchymal cell types of the liver [11,60].

The drawbacks associated with enzymatic tissue dissociation motivated the development of technologies to in vivo tag the nuclei surface based on genetically modified animals (often CRE/loxP-based) (Figure 2B). These strategies offer gentle and efficient enrichment of nuclei from a given cell population using affinity purification and include INTACT (isolation of nuclei tagged in specific cell types) [61] and NuTRAP (nuclear tagging and translating ribosome affinity purification) [62] (Figure 2B). The strategies also allow for FACS-based isolation of labelled nuclei (Figure 2C). Similarly, genetic approaches that use fluorescence labelling allow FACS of specific cells, nuclei, or both from intact liver tissue (Figure 2B). Examples include the purification of KCs and nuclei expressing NLS-tdTomato and the isolation of HSCs expressing YFP [11,12,63]. It is important to consider that the enriched cell type fraction is less pure in affinity purification-based approaches versus FACS-based approaches, possibly due to unspecific bead-pulldown events arising from clumping nuclei or the presence of ambient RNA and chromatin that is more thoroughly washed away during the FACS protocol/procedure. As with FACS, pre-knowledge of the target cell type is required to enable cell type-specific protein tag expression, making both approaches less suitable for the identification of unknown cell types.

## 4. The Cis-Regulatory Genome in Specific Cell Populations of the Liver

A few recent studies have used different genetic-based cell/nucleus enrichment strategies to uncover cis-regulatory regions in different cell types of the liver. These include specific studies of mouse hepatocytes, KCs, macrophages, and HSCs [11,12,55]. Each study used very different disease models to study liver tissue remodeling, which should be kept in mind when these studies are compared and related to the bulk liver analysis described above. Specifically, the hepatocytes were isolated from a mouse model replicating NASH induced by diets promoting steatosis and fibrosis, but without DIO [55]. KCs/macrophages were enriched from a rodent DIO model developing NASH [12], whereas the HSCs were isolated from mice treated with CCl_4_, which promotes NASH-like fibrosis without steatosis [11]. In the following section we describe the specific cell/nucleus enrichment strategies and methods applied to investigate the cistrome of specific cell types of the liver.

### 4.1. Hepatocytes

The relative increased percentage of non-hepatocyte cell types in the diseased, compared to healthy, liver may mask the detection of hepatocyte-specific processes in whole-tissue studies and lead to false conclusions if observed changes originate from infiltrating cell types. Thus, isolating hepatocytes in samples from NASH mouse models may lead to greater insights into hepatocyte regulatory plasticity in the disease. Recently, Loft and colleagues used the INTACT approach to tag and isolate hepatocyte nuclei in a NASH mouse model and performed ATAC-seq and RNA-seq to investigate gene-regulatory mechanisms during diet-induced steatosis and fibrosis [55]. ATAC-seq analysis of hepatocyte nuclei showed extensive chromatin remodeling during NASH progression (thousands of regions were dynamically regulated), which to some extent contradicts DNase-seq experiments from whole liver tissue isolated from DIO mice [23]. DNA motif analysis of the regulated cistrome predicted several TFs to be involved [55]. For example, motifs predicted to gain activity in the cis-regulatory regions included AP-1 motifs, which have previously been reported to facilitate progression of NAFLD [64]. In addition, motifs bound by GLIS2, EHF, and ELF3 were shown to be enriched in cis-regulatory regions associated with NASH, which agrees with bulk liver analysis [28]. Moreover, Elf3 and Glis2 expression was found to be upregulated in data mined from several diet-induced NASH mouse models and in humans [55,65,66,67,68,69,70,71]. Importantly, the knock down (KD) of Elf3 and Glis2 in mice challenged with NASH inducing diet, showed decreased inflammation, fibrosis, and apoptosis, but no differences in steatosis, suggesting a specific function for the development of fibrosis. Interestingly, KD restored hepatocyte-specific gene expression suppressed by NASH, suggesting normalization of hepatocyte function. A similar pattern was observed for specific non-parenchymal genes induced by NASH, indicating normalized gene expression of other cell types as well [55].

Motifs exhibiting decreased activity in NASH included HNF4α, HNF6, and surprisingly PPARα. HNF6 was previously reported to be enriched in cis-regulatory regions losing H3K27Ac in livers from DIO mice [28]. HNF4α was indicated to gain activity in some bulk liver studies [23,26], discussed above; however, the downregulation of HNF4α motif activity reported in hepatocytes may be explained by the NASH model or by the different methodology used for analysis: bulk liver tissue versus isolated hepatocytes. Importantly, HNF4 and HNF6 are known regulators of the hepatocyte-specific gene expression shown to be downregulated during NASH [72,73,74], suggesting that NASH remodels activity of linage determining TFs in hepatocytes. In contrast to reduced PPARα motif activity reported in hepatocytes, bulk liver analysis suggests increased PPARα motif activity in NAFLD [29]. This discrepancy could be an effect of the diets used to induce NASH or the timing of liver isolation during a circadian rhythm.

### 4.2. Kupffer Cells and Infiltrating Macrophages

Seidman and colleagues assessed the cis-regulatory landscape of resident KCs in NASH by isolating the KCs from DIO mice with a NASH phenotype [12]. As with the above-mentioned INTACT approach, this study used mice expressing a fluorescent marker (TdTomato-NLS) in nuclei of KCs, allowing FACS purification of KC whole cells and nuclei [75]. Chromatin accessibility was assessed by ATAC-seq in KCs isolated from the liver, and H3K27Ac was quantified using crosslinked nuclei that were FACS sorted for TdTomato. Remarkably, ATAC-seq analysis found minor changes to the chromatin accessibility in response to NASH, with fewer than 500 regions being significantly altered, which interestingly contradicts the more dramatic remodeling shown in isolated hepatocytes [55]. Genomic regions gaining accessibility in NASH were enriched for AP-1, ATF, and EGR2 motifs, while sites losing accessibility were enriched for PU.1 and SpiC motifs implicated in macrophage identity [76]. Assessing changes in H3K27Ac revealed more dramatic changes and identified close to 8000 cis-regulatory regions exhibiting significant changes in activity. Regions gaining activity were enriched for AP-1, ATF, NFAT, RUNX, and EGR motifs, whereas regions losing activity were enriched for LXR, MAF, and IRF motifs, implicated in lineage determination of KCs [76,77]. Moreover, 24% of the cis-regulatory regions associated with KC identity, exhibited significant loss of H3K27Ac upon NASH, indicating an overall suppressive effect of NASH on the regulatory regions defining KC identity, in agreement with KC identity genes being downregulated. Interestingly, assessing TF occupancy dynamics by ChIP-seq revealed both lost and gained LXR binding sites in NASH, suggesting that NASH leads to redistribution of LXRα occupancy. Additional functional analysis suggests that dynamic occupancy of ATF3 and reduced occupancy of SpiC in NASH is an underlying mechanism controlling the genomic redistribution of LXRα [12]. These findings highlight an interesting and complex feature of TFs, where one TF may act in several divergent gene regulatory networks depending on interactions with other factors [78].

### 4.3. Hepatic Stellate Cells

A recent study by Liu and colleagues profiled H3K4Me2, H3K27Ac, and gene expression in HSCs from CCl_4_-treated mice simulating liver injury seen in NASH [11]. They use autofluorescence from Vitamin A as a general marker for HSCs and a genetic approach to fluorescently tag (YFP) activated HSCs (aHSCs) during the treatment as well as inactivated HSCs (iHSCs) at two timepoints after the cessation of CCl_4_ treatment. HSCs were isolated using enzymatic tissue dissociation followed by FACS to separate quiescent HSCs (qHSC), aHSCs, and iHSCs. They found approximately 9190 H3K4Me2 and 18,800 H3K27Ac sites to be dynamically regulated between activation states of HSCs. TFs enriched in cis-regulatory regions associated with the quiescent or inactivated states generally belong to the same families and include ETS1/2 and GATA4/6. Functional analysis indicated that these TFs control the lineage commitment of HSCs, suggesting that the HSC activation triggered by hepatic injury leads to the inactivation of cis-regulatory sites associated with HSC identity. In agreement with these findings, the expression of ETS1 and ETS2 has previously been shown to be downregulated in fibrosis induced by DIO- or CCl_4_-induced NASH and in vitro during HSC activation [6]. In contrast, AP-1, TEAD, and NFêB motifs were enriched in cis-regulatory regions associated with the activated state. In agreement, AP-1 and NFkB play a functional role in the activation of HSCs [79,80,81,82]. Moreover, TEAD is known to be a major cofactor alongside YAP1 [83] and the pharmacological disruption of the YAP–TEAD complex attenuates fibrosis [84,85]. Thus, as HSCs are activated, the cis-regulatory landscape is remodeled by the loss of lineage determining TFs combined with the expression/activation of TFs induced by the altered intracellular environment of the injured liver.

### 4.4. NAFLD/NASH Remodel Lineage-Determining Cis-Regulatory Regions

Collectively, these cell type-resolved studies suggest that liver injury and NASH lead to a dramatic remodeling of the cis-regulatory regions of the genome, in agreement with the observations made from bulk liver analysis. Remarkably, for all studied cell types, it is observed that the cis-regulatory regions bound by lineage-specific TFs are reduced in accessibility, activity, or both, suggesting that cells change their functional states by disruption of their normal gene regulatory networks (Figure 3). In the pathophysiological condition, a new set of TFs operates the remodeled cis-regulatory genome leading to altered gene expression and new pathophysiological cellular states. For all analyzed cell types, this seems to involve TFs binding to the AP-1 motif, such as JUN, JUNB, FOS, FOSL2, suggesting that the upstream signaling pathways activating these TFs may be generally regulated by the hepatic microenvironment in NASH.

## 5. Emerging Technologies to Analyze the Cis-Regulatory Genome at Single-Cell Level

Categorizing cells in cell type-defined boxes is a simplification, as huge heterogeneity (cellular states) exists within given cell type populations in the healthy and diseased liver. For example, hepatocytes show distinct biological function according to their spatial localization in the liver lobule, described as liver zonation, driven by various biochemical gradients along the liver sinusoids such as oxygen, Wnt, hormones, and metabolites [86]. Moreover, LSECs are subject to distinct transcriptomic changes according to their localization in cirrhotic livers [87], and various macrophage subtypes have been identified in the context of NASH including lipid-associated macrophages (LAMs), c-LAMs, and embryonic or monocyte-derived KCs [4,88,89]. The development of laboratory and bioinformatic techniques, aiming to study the great diversity of cellular states, have exploded throughout the past decade, with spatial and single-cell transcriptomics being central breakthroughs. This branch of high-throughput methods has provided the field with valuable insight into cell type gene expression dynamics, intercellular communication, and tissue composition in NAFLD [60]. Importantly, single-cell RNA-seq (scRNA-seq) profiling does not require any biological pre-knowledge as individual cells are characterized independently of cellular tagging, which has facilitated the detection of novel and rare cell types across tissues [90,91,92]. Single cells can be profiled either as whole cells or nuclei (Figure 2A), and the different source of bias this might lead to (discussed above) also manifests here, emphasizing the importance of careful consideration of the experimental approach prior to conducting the analysis [58]. A major current limitation in the NAFLD/NASH single-cell field is that cells are characterized predominantly at the RNA level, and future development of single-cell epigenetic technologies will provide the field with a much deeper insight into disease mechanisms, including those in rare cell types that previously have proved difficult to isolate in sufficient quantities for cistromic analysis such as biliary progenitor cells and cholangiocytes whose isolation is further challenged by their many shared marker genes. The unique insight such techniques can provide in addition to RNA-based analyses has been discussed well by Shema et al. [93]. In general, single-cell transcriptomics have been reviewed extensively in recent years [94,95,96,97], and we refer to these for further insight.

### 5.1. Mapping Chromatin Accessibility at Single-Cell Resolution

Since ATAC-seq was introduced by Buenrostro et al. in 2013, it has been the method of choice for the mapping of accessible chromatin regions, as it entails several technical improvements in comparison to similar technologies such as DNase-seq [98]. Importantly, this has made its workflow adaptable to encompass single-cell profiling through several different strategies [99,100]. In particular, the commercialized droplet-based platform developed by 10× Genomics has gained popularity given that it meets the needs for cell throughput, capture efficiency, flexibility, and costs. Technically, single-cell ATAC-seq (scATAC-seq) is challenged by the low number of genomic target sites per cell and incomplete Tn5 cutting efficiency, leading to sparse, but also highly pure, output datasets. To meet the bioinformatic challenges associated with data sparsity, dedicated scATAC-seq analysis tools such as Cicero, SnapATAC, ArchR and Signac have been published recently, facilitating impressive insight into many of the relevant layers of biology for the detangling of gene regulatory networks [101,102,103,104]. For example, gene transcription can be predicted from the degree of chromatin accessibility in the gene-body and nearby cis-regulatory regions; the dynamics of cis-regulatory activity and cooperation can be studied by the changes in accessibility, the connections between cis-regulatory regions and genes, and by co-accessible regions in single-nuclei; TF candidates can be explored through motif enrichment or foot-printing analysis. As such, scRNA-seq and scATAC-seq overlap in some respects regarding insights into gene expression, but scATAC-seq additionally provides insight into the regulatory layer beneath. Exemplifying the strength of the method, recent studies utilized scATAC-seq in combination with scRNA-seq to identify gene regulatory networks in the brain [105,106]. Interestingly, Tedesco et al. recently introduced their single-cell genome and epigenome by using the transposases (scGET-seq) method that probes both open and closed chromatin in single cells utilizing the conventional Tn5 alongside a chimeric transposase with a chromodomain of a H3K9me3- (heterochromatin mark) binding protein. This strategy allows close evaluation of chromatin remodeling dynamics by also collecting information from the silenced regulome [107]. To our knowledge, none of these methodologies have been applied in a NAFLD context yet, but we foresee their great potential to identity novel cis-regulatory regions and master regulators driving pathogenic cellular behavior in the diseased liver.

### 5.2. Mapping Histone Modifications and TF Binding at Single-Cell Resolution

For many years, ChIP-seq has been the standard technique for the mapping of TF binding and histone modification, and its workflow includes chromatin fragmentation, immunoprecipitation (IP) of fixated DNA-protein interactions, and the sequencing of enriched DNA. Unfortunately, unspecific antibody binding and fixation-related epitope masking compromise data quality and a cell input count into the millions is required to reach an optimal signal-to-noise ratio. In a single-cell context, these technical limitations are even more pronounced because very few epitope targets must overcome the noise. A few studies, from the early single-cell era, approached this issue by adding cell-specific barcodes to chromatin fragments followed by IP on pooled chromatin for the analysis of histone marks [108,109]. Nonetheless, these ChIP-based strategies depend on the availability of highly specific antibodies and presumably do not enable the detection of fluctuating and less abundant TF sites.

A novel branch of enzyme-tethering technologies, including Cleavage Under Targets and Release Using Nuclease (CUT&RUN) and Cleavage Under Targets and Tagmentation (CUT&Tag), escape chromatin fixation and IP requirements, resulting in an extremely improved sensitivity compared with ChIP-seq [110,111,112]. The latter utilizes Tn5 coupled with Protein A (pA-Tn5), enabling antibody-directed transport of the enzyme to genomic targets followed by cleavage of the surrounding DNA and simultaneous insertion of sequencing adaptors into released DNA fragments. Importantly, the developers successfully adapted this strategy to a single-cell setup by subjecting tagmented single cells to nano-well barcoding prior to sequencing. Recently, Gopalan et al. introduced multi-CUT&Tag that supports profiling of multiple chromatin epitopes in single cells by loading the pA-Tn5 enzyme with antibody-specific barcodes [113]. This strategy has the potential to enable closer evaluation of the cooperation between different regulatory routes in the orchestration of a gene expression response. Importantly, this study as well as a study by Bartosovic et al. combine single-cell CUT&Tag (scCUT&Tag) with the 10× Genomics platform, paving the way for a more user-friendly and higher-throughput scCUT&Tag workflows in the future [114]. The scCUT&Tag studies highlighted above focus their analyses on histone marks or RNAPII activity, and only achieve TF mapping in bulk. However, a recent study has shown that TF occupancy can be mapped at a single-cell level [114], and we foresee that many more scCUT&Tag strategies and dedicated computational tools supporting the analysis of the complex datasets will follow soon. The future application of scCut&Tag-based technologies in a NAFLD context will provide the field with valuable insight into cell type specific gene regulatory networks and allow direct validation of putative TFs predicted in scATAC-seq studies.

### 5.3. Mapping Chromatin Interactions at Single-Cell Resolution

Although enhancer–promoter interactions can be predicted computationally from integrated scATAC-seq and scRNA-seq analyses [103], Hi-C provides direct proof of physical interactions between genomic sites, although conventionally it achieves this by averaging signals from millions of cells. In Hi-C, interacting chromatin regions are crosslinked, this is followed by DNA fragmentation and the addition of biotinylated nucleotides to 5′ ends. Next, interacting DNA regions are ligated together such that fragment junctions are tagged with biotin allowing a streptavidin-based pulldown and the sequencing of chimeric DNA fragments. In recent years, efforts to perform Hi-C at single-cell resolution [115,116,117,118,119,120], have been challenged by low coverage and low resolution of interactions, sparsity, complex laboratory workflows, high costs, and low throughput. More recently, Ramani et al. published an optimized version of their Sci-Hi-C strategy [121], which is based on the concept of combinatorial indexing. Thus, their protocol includes two barcoding steps performed on sets of either 25–100 K or ~25 nuclei with the rationale that barcode combinations will be unique to single cells allowing a streptavidin-based pulldown of interacting DNA fragments from pooled chromatin samples. Taken together, and thoroughly discussed in a recent review by Galitsyna and Gelfand [122], single-cell Hi-C is still in its infancy and optimized strategies for capturing DNA interactions will presumably be published in the coming years alongside dedicated computational analysis tools [123,124,125,126]. Future developments may allow deeper insights into genomic interactions in hepatic cell types during NASH progression and the assessment of scATAC-seq predictions through direct measurements.

### 5.4. Multi-Omics Technologies for the Profiling of Different Regulatory Layers

With the increasing repertoire of single-cell omics technologies, the next promising direction in the field is the simultaneous evaluation of different regulatory layers within the same cell. However, existing multi-omics technologies are, to our knowledge, yet to be applied to the liver. Already established, 10× Genomics offer their Chromium Single-cell Multiome ATAC + Gene Expression solution for the simultaneous profiling of chromatin accessibility and gene transcription, enabling confident predictions of the interactions between cis-regulatory sites and target genes. Other 10× Genomics compatible technologies include cellular indexing of transcriptomes and epitopes by sequencing (CITE-seq) and RNA expression and protein sequencing assay (REAP-seq) that profile gene transcription and protein expression by utilizing oligo-tethered antibodies for sequencing-based protein identification [127,128], thereby allowing the cellular state, captured at the RNA level, to be analyzed in association with cellular function captured at the protein level. In a plate-based approach, Protein-indexed (Pi)-ATAC detects protein abundance and chromatin accessibility in individual cells, allowing the integrated analysis of TF expression and motif activity [129]. Importantly, this can exclude false–positive TFs and improve the analysis of post-translationally regulated TFs. However, protein detection is based on antibody labelling and FACS, limiting the number of analyzed proteins. Still in preprint, Chen et al. recently introduced Nuclear protein Epitope, chromatin Accessibility, and Transcriptome sequencing (NEAT-seq) for the simultaneous detection of nuclear proteins, chromatin accessibility, and gene transcription, facilitating impressive insight into the regulatory link between TF binding at accessible cis-regulatory sites and induced gene transcription [130]. Unfortunately, all protein-including studies highlighted above look solely at protein presence/abundance and do not provide a direct measurement of genome binding events of TFs. Thus, we look forward to the further maturing of CUT&Tag technology (or similar) so that this central aspect of transcriptional gene regulation can be explored in a multi-omics setup. Taken together, multi-omics approaches add an extra layer of resolution to the analysis by addressing the regulatory heterogeneity across cells.

### 5.5. Linking Gene Expression to Cellular Function: Single-Cell Proteomics and Single-Cell Metabolomics

Transcriptional gene regulation represents only one aspect of a cell’s regulatory machinery and molecular events acting at the proteomics and metabolomics level, for example, also take part in the narrow control of biological processes carried out by cells. Accordingly, great efforts are being invested in the development of single-cell techniques allowing insight into these additional layers of biology. The current progress and challenges associated with single-cell proteomics or single-cell metabolomics have been outlined very recently [131,132]. It is worth highlighting that Rappez and colleagues applied their single-cell metabolomics technology SpaceM in an in vitro NAFL/NASH model and detected a steatosis-prone hepatocyte subpopulation (24% of all cells) in the milder NAFL setting, which expanded to 93% of all cells in an inflamed context mimicking NASH [133]. It will be exciting to investigate whether such hepatocyte subpopulation exists in vivo, representing a potential novel drug target for early disease reversal. Taken together, future integration of a wide spectrum of data—ranging from epigenomics, transcriptomics, proteomics, and metabolomics—will enable unprecedentedly detailed studies of the regulatory mechanisms that drive cellular behavior in health and disease.

## 6. Summary and Perspectives

Early genomics studies of intact liver tissue have provided some general insights into the cis-regulatory landscape involved in the development of NAFLD. Based on this, some TFs are predicted to have functional importance, and a few have been validated experimentally; however, these bulk liver-based studies are limited by their lack of cellular resolution. Recent studies using cell/nucleus enrichment strategies have provided additional insights into specific cell populations of the liver and thus expanding our understanding of the cis-regulatory landscape involved in NASH. The development of single-cell omics technologies now enables unprecedented insights into the cis-regulatory genome in every cell type of the liver. This not only allows in-depth analysis of NAFLD/NASH disease models but also provides a unique opportunity to map the cis-regulatory genome of liver biopsies from patients diagnosed with NAFLD or NASH. These gene regulatory maps can be combined with genomic information from individual patients, and thus give an opportunity to link mutations to disease progression and potentially to personalize medical treatment using future drugs against NAFLD/NASH.

## Figures and Tables

**Figure 1 cells-11-00870-f001:**
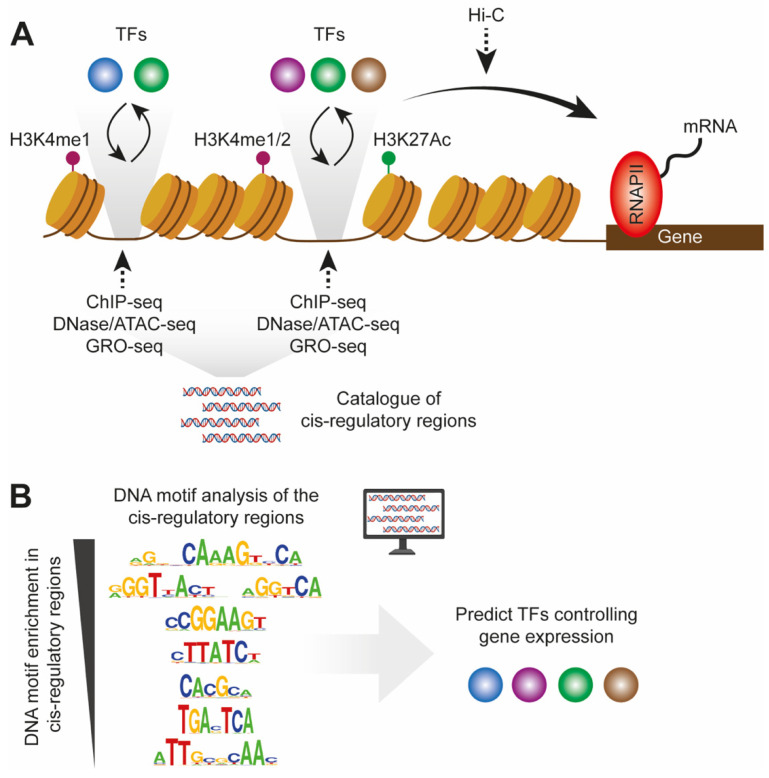
Identification of cis-regulatory regions in the genome and prediction of bound transcription factors. (**A**) Cis-regulatory regions of the genome are characterized by relatively high accessibility (probed by ATAC-seq or DNase-seq), with nucleosomes nearby with H3 mono- or dimethylated at lysine 4, acetylated at lysine 27 (probed by ChIP-seq), or both, or they are characterized by the expression of eRNAs (probed by GRO-seq), or by both. Based on the -seq methodologies, the cis-regulatory genome of a given cell population can be isolated and analyzed bioinformatically. (**B**) DNA sequence analysis of the cis-regulatory genome can reveal enriched sequence motifs bound by specific TFs. Consequently, the analysis of cis-regulatory genomes from different cellular states can provide information on TFs controlling gene expression in these cellular states.

**Figure 2 cells-11-00870-f002:**
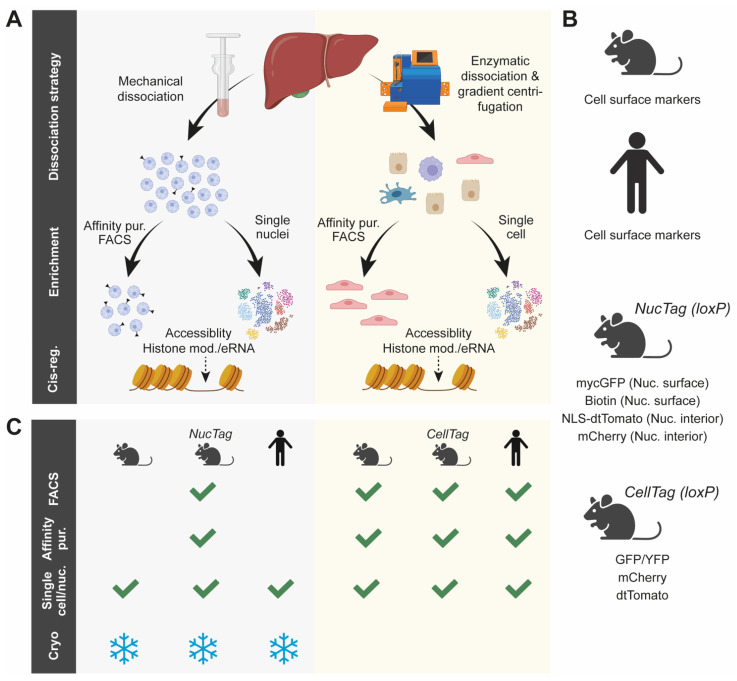
Methodologies for isolating cells and nuclei from liver tissue applicable for cistromic analysis. (**A**) Liver tissue can be dissociated mechanically (**left**) or enzymatically (**right**) to release nuclei or cells, respectively. Cells or nuclei from specific cell populations can subsequently be purified using affinity purification or FACS. Alternatively, cells or nuclei can be studied by single-cell resolved analysis of the cistrome. (**B**) Based on cell surface markers, specific cell populations can be isolated from the livers of rodent NAFLD/NASH disease models or the liver biopsies from human donors. Alternatively, specific nuclei or cells can be labelled by fluorescent proteins or biotin (often CRE/loxP-based). If the nuclei are labelled at the surface, this enables affinity purification. If labelled in the nuclei interior, FACS can be used for enrichment. Similarly, cells can be labelled by the cell specific expression of fluorescent proteins. (**C**) Cells and nuclei isolated from human biopsies or different rodent cell/nucleus tagging models can be purified by different strategies. Cells isolated by specific cell surface markers or cell specific labelling strategies can be purified by FACS, affinity purification, or both. In contrast, cell-specific nuclei can only be purified from genetic models that specifically label the nuclei. Both cells and nuclei isolated from liver can be subjected to single-cell/nucleus analysis and only nuclei can be recovered from cryopreserved samples.

**Figure 3 cells-11-00870-f003:**
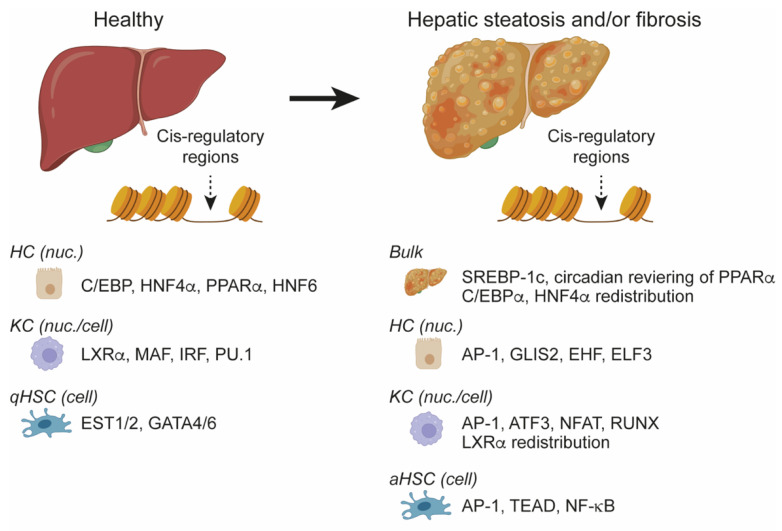
Dynamic regulation of the cis-regulatory genome of hepatic cells during development of hepatic steatosis, fibrosis, or both. A cistromic analysis of bulk liver and several different isolated cell types has revealed several important TF motifs and TFs involved in the processes. HC: Hepatocyte, KC: Kupffer cell, HSC: Hepatic stellate cell. The indicated TFs were identified based on a cistromic analysis of nuclei (nuc.), cells, or the whole liver (referred to as bulk).

## Data Availability

Not applicable.

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
