# Peer review of "Cell-Type Resolved Insights into the Cis-Regulatory Genome of NAFLD"

_cells, 2022, doi:10.3390/cells11050870_

Round 1

Reviewer 1 Report

The authors should discuss the role and contribution of cholangiocytes that have been shown to play a role in NAFLD/NASH with regards to ductular reaction and inflammation. This section should be added to the review. 

Author Response

We thank the reviewer for highlighting this important aspect of NAFLD.  To our knowledge, cis-regulatory mechanisms in cholangiocytes have yet to be studied in the context of NAFLD. In our revised version of the review, we now mention the ductular reaction aspect of NAFLD in the introduction. And later in section 5 we discuss that emerging single cell epigenetic technologies will leverage cistromic analysis of rare cell types such as cholangiocytes for future analysis.  

Reviewer 2 Report

The manuscript summarizes the available information from the literature about the cis-regulatory landscape associated with NAFLD. It addresses an interesting and important part of the many regulatory layers of the disease. The review provides a comprehensive description of the current knowledge on chromatin structure changes and also the numerous experimental procedures developed in the recent years.

The text is easily read. The Figures are clear and help to understand the concepts and the logic of the experimental approaches.

Author Response

We thank the reviewer for the positive feedback to the manuscript.

Reviewer 3 Report

The review presented by Dam TV, Toft N.I. and Grontved L tries to collect many of the new insights into the gene regulatory mechanisms associated during the pathogenesis of non-alcoholic fatty liver disease and NASH specifically focusing on cis-regulatory regions and mechanisms. The need for the study in individual cell populations of the liver is precisely discusses and many of the studies carried out at this level are described specifically in this pathology. Comprehensive and well-updated bibliography. Delve into emerging technologies addressing to the analysis of the cis-regulatory genome at single cell level and offers a very interesting and necessary perspective to go deeper into all the mechanisms that specifically govern the development of the disease. It is innovative and provides a very good and novel vision, well-made illustrative figures. Only minor changes are suggested:

  • The title refers to a vision that is too general for what later turns out to be the review. In this case, rather than a description of the epigenetic mechanisms that govern the development of NAFLD-NASH, it would be interesting to emphasize that the study focuses on single cell approaches and their technologies. Not much epigenetic is detailed in the manuscript, rather it refers to certain marks or processes of chromatin remodeling that help control and regulate gene expression. Perhaps a less general and more groundbreaking title could be interesting.
  • Certain terminologies such as eRNAs and GRO-seq are not well defined.
  • In the last section that refers to multi-omics technologies, there is a very interesting work that talks about SpaceM, an open-source method for in situ single-cell metabolomics that can contribute to offer a new perspective of analysis in this field. (Nat Methods 2021 PMID: 34226721)

Author Response

We thank the reviewer for pointing out that the title is rather generic, non-descriptive, and slightly misleading considering the content in this review. We altered the title to better match the focus and content of the review (see revised manuscript). Also, we appreciate the suggestion to include a discussion of the SpaceM technology. In our revised version of the manuscript, we have added a new section 5.5 where we briefly outline the progress and challenges associated with scMetabolomics as well as scProteomics and we highlight SpaceM technology in this section. Moreover, in the revised version we have elaborated a bit on how GRO-seq and eRNAs can be used as a measurement of enhancer activity (section 2).